# Predicting the target landscape of kinase inhibitors using 3D convolutional neural networks

Georgi K. Kanev[1,2], Yaran Zhang[2], Albert J. Kooistra [1,3], Andreas Bender[4], Rob Leurs[1], David Bailey[5,6], Thomas Würdinger[2,5], Chris de Graaf[1¤], Iwan J. P. de Esch[1], Bart A. Westerman [2,5] *

1 Division of Medicinal Chemistry, Amsterdam Institute of Molecular and Life Sciences (AIMMS), Vrije Universiteit Amsterdam, Amsterdam, The Netherlands, 2 Department of Neurosurgery, Amsterdam University Medical Centers, Cancer Center Amsterdam, Brain Tumor Center Amsterdam, Amsterdam, The Netherlands, 3 Department of Drug Design and Pharmacology, University of Copenhagen, Copenhagen, Denmark, 4 Centre for Molecular Science Informatics, Department of Chemistry, University of Cambridge, Cambridge, United Kingdom, 5 The WINDOW consortium, www.window-consortium.org, 6 IOTA Pharmaceuticals Ltd, St Johns Innovation Centre, Cambridge, United Kingdom

¤ Current address: Heptares Therapeutics, Steinmetz Building, Granta Park, Great Abington, Cambridge, United Kingdom
* a.westerman@amsterdamumc.nl

**Data Availability Statement:** All data and scripts will be provided through a github link https://github.com/bartwesterman/Kanev_et_al_2023.

## Abstract

Many therapies in clinical trials are based on single drug-single target relationships. To further extend this concept to multi-target approaches using multi-targeted drugs, we developed a machine learning pipeline to unravel the target landscape of kinase inhibitors. This pipeline, which we call 3D-KINEssence, uses a new type of protein fingerprints (3D FP) based on the structure of kinases generated through a 3D convolutional neural network (3D-CNN). These 3D-CNN kinase fingerprints were matched to molecular Morgan fingerprints to predict the targets of each respective kinase inhibitor based on available bioactivity data. The performance of the pipeline was evaluated on two test sets: a sparse drug-target set where each drug is matched in most cases to a single target and also on a densely-covered drug-target set where each drug is matched to most if not all targets. This latter set is more challenging to train, given its non-exclusive character. Our model's root-mean-square error (RMSE) based on the two datasets was 0.68 and 0.8, respectively. These results indicate that 3D FP can predict the target landscape of kinase inhibitors at around 0.8 log units of bioactivity. Our strategy can be utilized in proteochemometric or chemogenomic workflows by consolidating the target landscape of kinase inhibitors.

## Author summary

In this publication we have set up a new method to predict the targets kinase inhibitors. This information is important since it allows to understand how the inhibition of multiple targets by an inhibitor translates to its potency. For this, we have used a convolutional neural networks (CNN) strategy that is commonly used to classify images with common

**Funding:** G.K., B.A.W. C. d. G., T.W., R.L. and I.d.E. are supported by Amsterdam Data Science (ADS), B.A.W., D.B. and T.W. are supported by the Brain Tumour Charity Grant 488097 (WINDOW consortium), B.A.W. is supported by the Innovation Exchange Amsterdam (IXA) grant APCA-PoC-2017 and G.K. is supported by the Cancer Center Amsterdam (GlioSPARK project 2018-2-19). The funders had no role in study design, data collection and analysis, decision to publish, or preparation of the manuscript. None of the funders played any role in the study design, data collection and analysis, decision to publish, or preparation of the manuscript.

but also different elements, for instance to recognize classes (predicting which animal is shown) or subtle changes (predicting which emotions are shown on a face). In this case we have used CNN models to detect patterns in 3-dimensional information from kinase structures and matched this to the bioactivity of their respective inhibitors. We show that our method has a similar performance as benchmark methods in the field. However, since recognized patterns can be made explicit, our method could help to make the underlying artificial intelligence method explainable. In the future this could lead to more optimal matching of multi-target drugs to diseases that have multiple vulnerabilities.

## 1 Introduction

The average cost of developing a new drug has increased by a factor of 9 in the last 50 years.[1] And although personalized therapies are steadily showing their value in clinical trials, [2,3] these trials can suffer from therapy resistance, [4–6] requiring adaptive treatment strategies. Network-based and proteochemometrics approaches [7–9] introduced a decade ago offer a paradigm shift from highly selective single-target drugs to multi-target drugs where the latter is more likely to achieve desired clinical efficacy and potentially lower the research and development (R&D) costs. [10,11] While kinase inhibitors are known to be promiscuous, [12,13] currently, most of them have been tested on up to two kinases in publically available databases and resources (**Fig 1A**). [14–18] A chemogenomics framework able to identify the target landscape of kinase inhibitors might facilitate the selection of highly tailored therapies and provide opportunities for effective therapies by, e.g., determining the polypharmacological profile of drugs. [12,19]

Currently, more than 70 small-molecule kinase inhibitors have been approved by Food and Drug Administration (FDA). [20] Most approved drugs compete with adenosine 5'-triphosphate (ATP) to bind to the ATP binding site of the kinase fold to prevent downstream signal transduction. [21] For comprehensive structural, sequence alterations, and inhibitor binding comparison of the protein kinases, the reader is referred to Kanev et al. [22] Due to the highly conserved kinase fold, most kinase inhibitors inhibit multiple targets, [20,23] commonly between three (e.g., Selumetinib) and 60 kinase proteins (e.g., Staurosporine, Dovitinib, Vandetanib, and Dasatinib). It is assumed that the activity profile across many protein kinases determines the therapeutic efficacy and safety of kinase inhibitors. [10,24–27]

Virtual screening (VS) methods [28] provide a significant reduction in costs and improvement of the hit rate to conventionally used high-throughput screenings (HTS) in the early stage of drug discovery. [29] These methods screen (large) libraries of small molecules for novel and diverse bioactive compounds. [30,31] Among the VS methods, machine learning (including quantitative structure-activity relationship (QSAR)) models provide the ability to learn patterns and use them to make predictions on unseen data. [32–35] Widely applied machine learning algorithms include random forests, [35–39] support vector machines (SVM), [9,40,41] and neural networks. [42–45] Neural networks [46] have a long history in predicting binding affinity of small molecules [42,43,45,47,48] with ever increasing popularity in drug discovery efforts. [49,50] In particular, their ability to handle data without the need for feature selection, [46,51] easy hyper-parameter optimization to increase performance, and good methods like dropout [52] to avoid overfitting proved useful in setting up machine learning workflows. [53]

More recently, convolutional neural networks (CNN) [46,54,55] have also been applied to predict the binding affinity of small molecules, [47,56–64] learn molecular fingerprints,

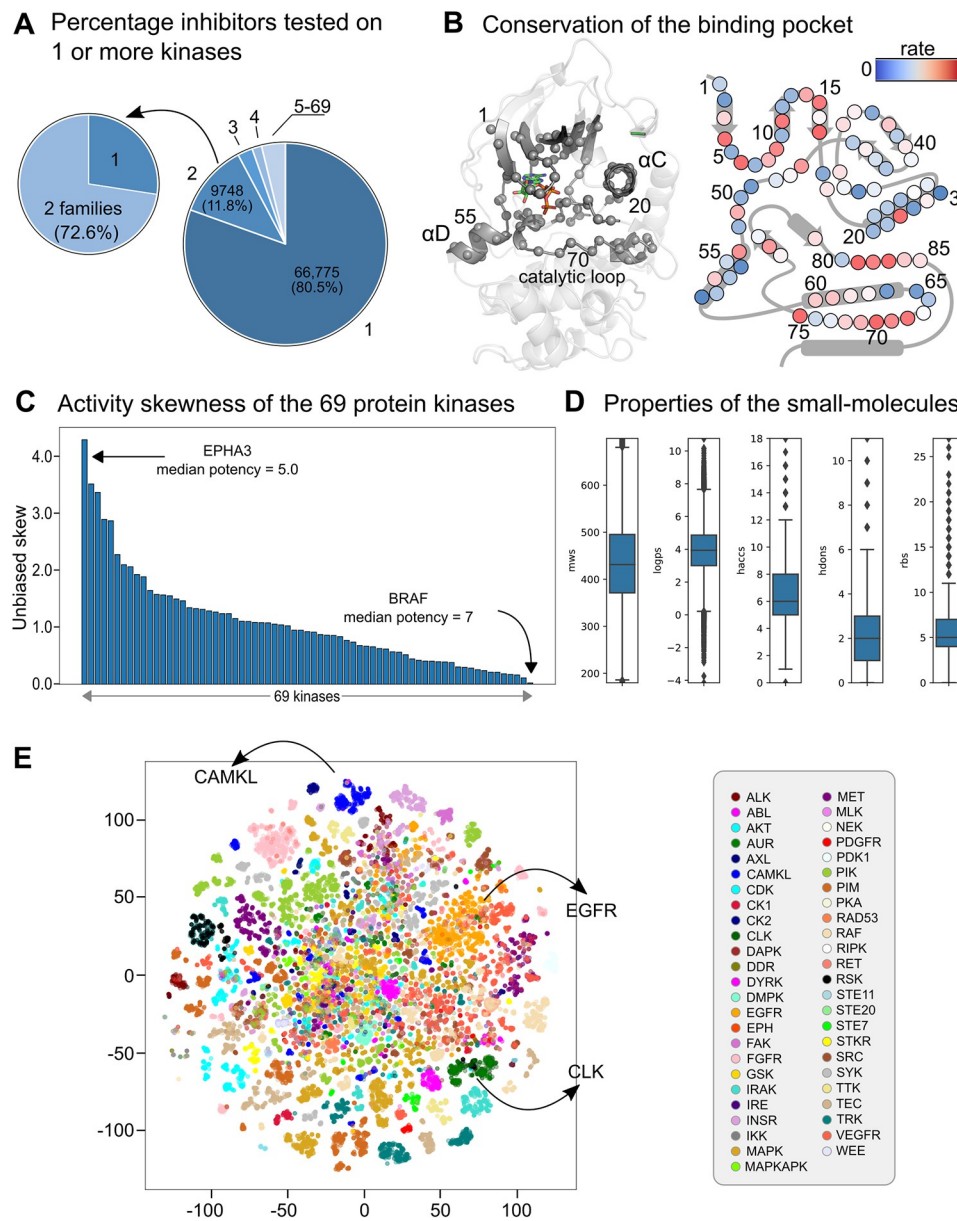

**Fig 1. The small molecule, kinase and activity overview of the integrated data. (A)** Piechart showing the percentage of inhibitors experimentally tested for one or more kinases. 80.5% of the inhibitors in the data set are tested for a single kinase, 11.8% for 2 kinases and ~8% for more than 2 kinases. In roughly 30% of the inhibitors tested for 2 kinases, the kinases belong to the same family. **(B)** Plot showing the conservation rate of the 85 amino acids of the binding pocket as defined in KLIFS. Examples of highly conserved motifs include the HRD motif (position 68–70) and the DFG motif (positions 81–83). **(C)** The unbiased activity skewness of the 69 kinases used in the machine learning pipeline. Kinases with lower skewness have more inhibitors with higher potency (-log[activity]) than kinases with higher skewness. The formula used to calculate the unbiased skew for a given kinase: skew $= \mathrm{E}\left[\left(\frac{X-\mu}{\sigma}\right)^3\right]$ where X is each activity value, μ is the mean, σ is the standard deviation, and E is the expectation operator. **(D)** Boxplots with properties of the small molecules. From left to right: molecular weight (MW), LogP, hydrogen-bond acceptors (H ACC), hydrogen-bond donors (H DON), and rotatable bonds (RBs). **(E)** t-SNE plot of the small molecules with activity ($IC_{50}$, Ki, or Kd) less than or equal to 100 nM. The plot was generated using the Morgan fingerprints of the compounds with a radius of 2 and bit length of 1024 and the tSNE package from scikit-learn 1.0.2 with parameters learning rate set to "auto" and PCA initialization. Each kinase family was assigned a unique color.

[59,65,66] detect chemical motifs, [67] and predict properties of small molecules [68] among others. This class of neural networks uses convolutional layers designed to take advantage of the 2D or 3D arrangement of the data and provides the ability to learn and detect local motifs across a predefined space. [46] In 2015, AtomNet was the first CNN to use 3D structural data to predict the binding affinity of small molecules. [57] Further advances in the field resulted in the application of 3D structural CNN to post-process molecular docking poses, [47,69,70] and predicted binding affinity using (mainly) PDBbind [71] for training. [56,58]. In 2020, Jocelyn Sunseri and David R. Koes released the libmolgrid library for gridding 3D structural data for convolutional neural networks. [72]

Here, we developed the machine learning pipeline 3D-KINEssence that 1) generates a new type of 3D structural fingerprints (3D FPs) using 3D convolutional neural network (3D-CNN) and 2) uses random forest (RF) models to predict kinase-inhibitor bioactivities (potency). 3D-CNN is trained on 3D biomolecular structural data, and its learned features are used to generate 3D FPs. The performance of the 3D FPs was estimated using large-scale protein kinase activity data and compared to commonly used protein features like the z-scales and ProtVec. Random forest models were utilized to evaluate the benefits of having protein features (versus only inhibitor features) and the performance of the different protein features in predicting potency. The evaluation was performed on two different test sets: a sparse and mostly mutually exclusive test set versus a dense, almost entirely covered chemogenomic test set (to evaluate better the performance of the protein features). We show that the newly generated 3D FPs outperform the commonly used z-scales and perform equally to the ProtVec protein features.

## 2 Results

### The known inhibitor-kinase activity space is very sparse

To determine the experimental inhibitor-target activity space, we collected bioactivity data from ChEMBL v31 [73] and Christmann-Franck et al. 2016 [38] for kinases with at least ten solved crystal structures (see materials and methods for the exact filtering steps). The collected set contained 169,723 activity data points comprising 69 unique protein kinases and 82,960 unique small molecules. The dataset is sparse, covering only 3.0% of all possible measurements. The 69 protein kinases represent all nine major protein kinase groups (**S1 Table**). Machine learning methods can be utilized to design drugs that achieve better clinical efficacy by predicting the missing potencies in this 82,960 by 69 matrix (and beyond) and identifying the targets and off-targets of drugs.

Sequence and structural alignments of the selected kinases reveal the highly conserved amino acids and motifs (G-rich loop, $K^{III.17}$, $E^{\alpha C.24}$, $N^{c.l.75}$, HRD motif, DFG motif) and their spatial positions in the binding site (**Fig 1B**). However, this plot also reveals that a large part of the binding site is less conserved on the sequence level, allowing binding pockets to appear/open or disappear/close. This information could be utilized by the convolutional neural network when trained on the 3D biomolecular structural data and, as a result, ends in the newly generated fingerprints (3D FP). The activity skewness plot of the kinases reveals that kinases generally have a positive skew (distribution is skewed toward the lower potencies) while a few are close to 0 (distribution is symmetrical) (**Fig 1C**). The median potency of BRAF (located to the extreme right of the distribution) equals 7.0, while the median potency of EPHA3 (located to the extreme left of the distribution) equals 5.0. The skewness differs significantly when calculated per test sets rather than the entire data set (**S1 Fig**). The dense set is skewed to the left (positive skew) as each compound is tested on at least 60 kinases. Selective compounds would inhibit not more than a few kinases; thus, most of these compounds will show low to no

activity on most kinases (low to no potency). The kurtosis plot (generated with Fisher's definition) is close to 0 for most kinases, with five kinases having a kurtosis of higher than 5 (**S2 Fig**). 80.5% of the small molecules are tested on a single kinase, 11.8% on two kinases, and just ~8% on more than two kinases (**Fig 1A**). Most (72.6%) of the two kinase profiles in the 11.8% subset belong to two families. The small molecules covered a large chemical space, with molecular weight (MW) ranging from 180 to 700, hydrogen bond acceptors (HB ACC) from 0 to 18, hydrogen bond donors (HB DON) from 0 to 11, rotatable bonds (RB) from 0 to 27 and LogP (using Crippen's approach[74]) between -4 and 10 (**Fig 1D**).

Interestingly, the t-distributed stochastic neighbor embedding (t-SNE) plot generated with Morgan fingerprints (radius 2, 1024 bits) reveals that some small molecule clusters preferentially inhibit specific kinase families. Most of the small molecules are located in the middle of the plot indicating a probable lack of distinctive features toward a specific family (**Fig 1E**). For example, the inhibitors of EGFR, CAMKL, and CLK form their own clusters.

## 3D convolutional layers allow extracting features from the molecular structures of protein kinases

Kinase structures were obtained from KLIFS [75,76] through the KNIME analytical platform. [77] These structures were used in the 3D-KINEssence pipeline to train a 3D Convolutional Neural Network (3D-CNN) capable of learning important structural features of individual protein kinases directly from their structures (**Fig 2**). The objective of 3D-CNN is to use the learned structural features to recognize the 69 kinases. In total, 3394 structures were used to train and test the 3D-CNN model (**S2 Table**). Structures were carefully prepared with Maestro before providing them as input to the libmolgrid library [72] for the generation of the 3D grids. The model was trained on 3044 structures and tested on 350 structures. The output of the flatten layer was used to generate the 3D convolutional fingerprints (3D FP) (**Fig 2**). This flatten layer "flattens" the learned representations of the convolutional layers (from 3D to 1D) while keeping spatial information. The cross-entropy loss of the CNN model was 0.02. The results indicate that the 3D convolutional neural network can learn sufficient information from the 3D structures of protein kinases, allowing it to separate the individual kinases accurately.

In addition to the 3D convolutional fingerprints, one-hot-encoding, z-scales, [78] and Prot-Vec [79] were also included in the analysis (**Fig 3A**). All of the protein features were tested in combination with Morgan fingerprints (models 2 to 6) generated with RDKit. [80] The Morgan fingerprints (compound features) were also tested independently (model 1) to evaluate the contribution of the different protein features. The z-scales were tested in 2 settings—per residue in the KLIFS binding site and the whole KLIFS binding site sequence (85 residues). In the case per residue, for each of the residues in a sequence, five z values were generated and used as input features (5 x 85 = 425), while in the case of a whole sequence, five z values were generated for the whole KLIFS binding site sequence.

## Protein features contribute to better scoring of machine learning workflows

Random forest (RF) models were built to evaluate the different protein features and their contribution alongside Morgan fingerprints. The compound features were generated with RDKit's Morgan fingerprints using the simplified molecular input line entry specification (SMILES). [81] The random forest models were tested on two test sets: sparse and densely-covered test sets (**Fig 3B**). The train and test sparse and dense data sets are generated using different split criteria from the collected bioactivity data from ChEMBL v31 [73] and Christmann-Franck

## 3D-KINEssence

### ① Generating 3D structural fingerprints

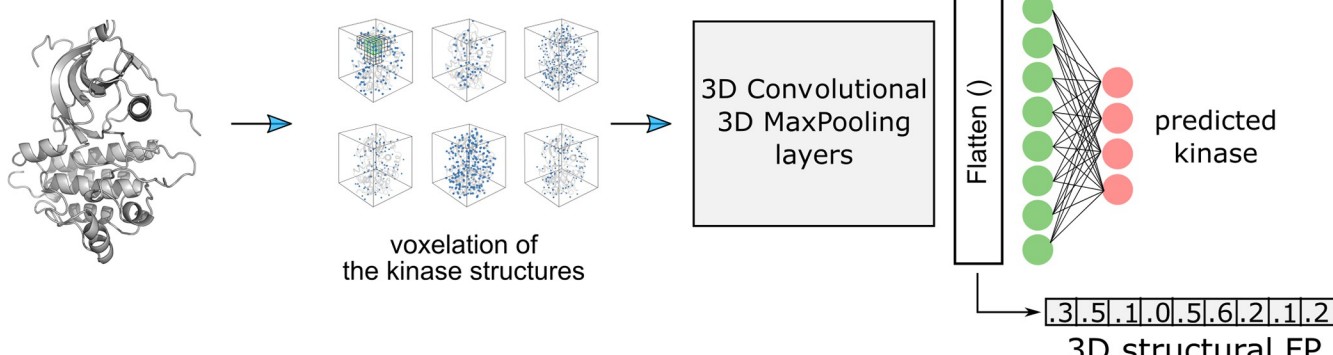

### ② Predicting inhibitor-kinase bioactivities

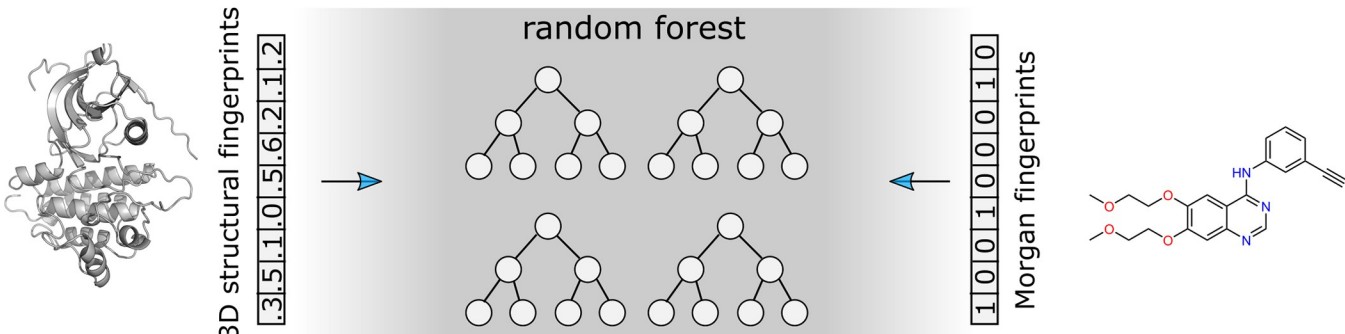

**Fig 2. Schematic representation of the 3D-KINEssence pipeline.** The structures of the 69 chosen kinases were obtained from KLIFS [69,70], prepared with Maestro, and gridded with libmolgrid before being provided to 3D convolutional neural networks to train. The features learned in the convolutional layers are flattened and used as 3D convolutional fingerprints (3D FP) in the random forest workflow. The newly generated 3D FP, along with Morgan fingerprints, were used as inputs to the random forest model to predict inhibitor-kinase potency. The representation of the voxels is adapted from Green et al. [85].

et al. 2016 [38]. While the sparse set is only 15% bigger than the dense set, its small molecules number is ~74 times more prominent (**Fig 3B**). The densely-covered test set represents a typical chemogenomics [1,82,83] or proteochemometrics [8,9] matrix where each small molecule is experimentally validated on at least 60 kinases (out of 69 in total). The structures of the 278 inhibitors of the dense set can be found in **S3 Table**.

Models trained with the same features show root-mean-square error (RMSE) differences that vary from 0.1 to ~0.7 depending on the test set they are evaluated on. For example, model 1 (only Morgan fingerprints) scored 1.46 on the dense and 0.8 on the sparse sets. Adding one-hot-encoded kinases features (model 2) lowered the RMSE with ~0.1 to 0.71 on the sparse set. The RMSE of model 3, evaluated on the sparse test set, was 0.7. Models 4, 5, and 6 (z-scales per residue (425), ProtVec, and 3D FP, respectively) scored very similar RMSEs: ~0.69. (**Fig 4A**). The $R^2$ scores for models 1 to 6 trained on the sparse data set range from 0.64 (model 1) to 0.74 (models 5 and 6). The $R^2$ scores of models 2, 3, and 4 were 0.719, 0.728, and 0.736, respectively. We have also evaluated how the RMSE and $R^2$ per kinase would change if the median potency for each kinase in the training set were taken and used as the predicted value

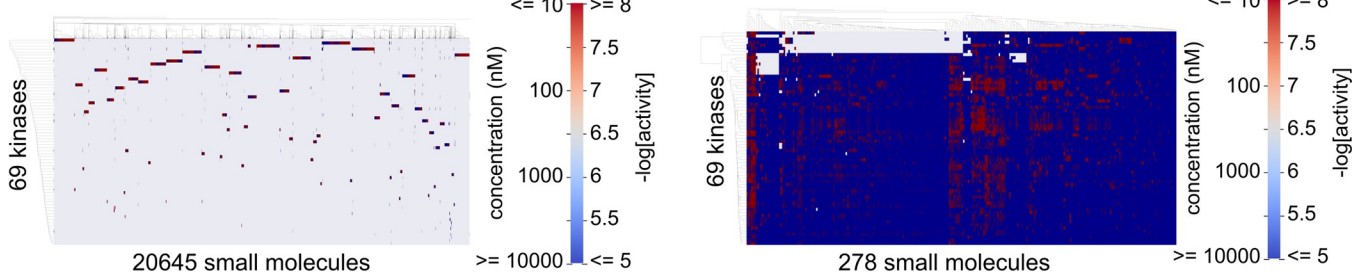

**Fig 3. The input features and the test data sets used in the machine learning pipeline. (A)** The input features used in the machine learning pipeline. Model 1 uses only the Morgan fingerprints from the rdkit package with a radius of 2 and length of 1024. In all other models, protein features and Morgan fingerprints were used. The kinases were one-hot-encoded for model 2. In model 3, the five z-scales were calculated using the 85 amino acid sequences of the binding pocket as defined in KLIFS. The z-scales in model 4 use the same sequences, but they were calculated per residue. The same sequences were used to generate the ProtVec features for model 5. Model 6 utilizes the 3D FP generated using a 3D Convolutional Neural Network. **(B)** The test sets are used to evaluate the machine learning pipelines. On the left is the sparse drug-target test set comprising 20645 small molecules and 69 kinases with a coverage of ~1.5%, and on the right is the densely covered test set comprising 278 small molecules and 69 kinases with a coverage of ~95%.

(constant) by the models for that kinase in the test sets (**S4 Table**). The median RMSE of all 69 RMSEs (one for each kinase) of the sparse test set is 2.287, indicating that using the median potency of all inhibitors of a kinase from the train set as a constant to predict the potencies for that kinase in the test set is not sufficient. The median $R^2$ of all 69 $R^2$ scores is -1.873 (**S4 Table**).

**A**

| | RMSE | | | | | | % predicted activities | | | |
|---|---|---|---|---|---|---|---|---|---|---|
| | RMSE | tanimoto 0.0 - 0.4 | tanimoto 0.4 - 0.6 | tanimoto 0.6 - 0.7 | tanimoto 0.7 - 0.8 | tanimoto 0.8 - 1.0 | 0 - 0.5 logs | 0.5 - 1.0 logs | 1.0 - 1.5 logs | 1.5 - ∞ logs |
| **Sparse drug-target test set** | | | | | | | | | | |
| model 1 mol fp | 0.80 | 1.01 | 1.06 | 0.93 | 0.84 | 0.70 | 52% | 29% | 12% | 7% |
| model 2 mol fp + one hot encoded | 0.71 | 0.95 | 1.02 | 0.86 | 0.76 | 0.59 | 58% | 27% | 10% | 5% |
| model 3 mol fp + z-scales protein (5) | 0.70 | 0.98 | 1.02 | 0.84 | 0.74 | 0.58 | 60% | 26% | 9% | 5% |
| model 4 mol fp + z-scales pocket (425) | 0.69 | 0.95 | 1.01 | 0.83 | 0.73 | 0.58 | 61% | 26% | 9% | 4% |
| model 5 mol fp + ProVec | 0.69 | 0.99 | 0.99 | 0.82 | 0.72 | 0.57 | 61% | 26% | 9% | 4% |
| model 6 mol fp + 3D FP | 0.68 | 0.96 | 1.01 | 0.82 | 0.72 | 0.57 | 61% | 26% | 9% | 4% |
| | | | | | | | | | | |
| **Densley-covered test set** | | | | | | | | | | |
| model 1 mol fp | 1.46 | 1.05 | 1.13 | 1.29 | 1.21 | 1.90 | 22% | 25% | 21% | 32% |
| model 2 mol fp + one hot encoded | 1.19 | 0.90 | 1.06 | 1.08 | 1.09 | 1.41 | 32% | 24% | 22% | 23% |
| model 3 mol fp + z-scales protein (5) | 0.91 | 0.78 | 0.83 | 0.90 | 0.88 | 1.01 | 35% | 37% | 19% | 8% |
| model 4 mol fp + z-scales pocket (425) | 0.79 | 0.75 | 0.75 | 0.76 | 0.77 | 0.84 | 44% | 37% | 14% | 5% |
| model 5 mol fp + ProVec | 0.79 | 0.75 | 0.73 | 0.78 | 0.77 | 0.86 | 44% | 37% | 14% | 5% |
| model 6 mol fp + 3D FP | 0.80 | 0.76 | 0.76 | 0.77 | 0.79 | 0.84 | 46% | 35% | 14% | 5% |

**B**

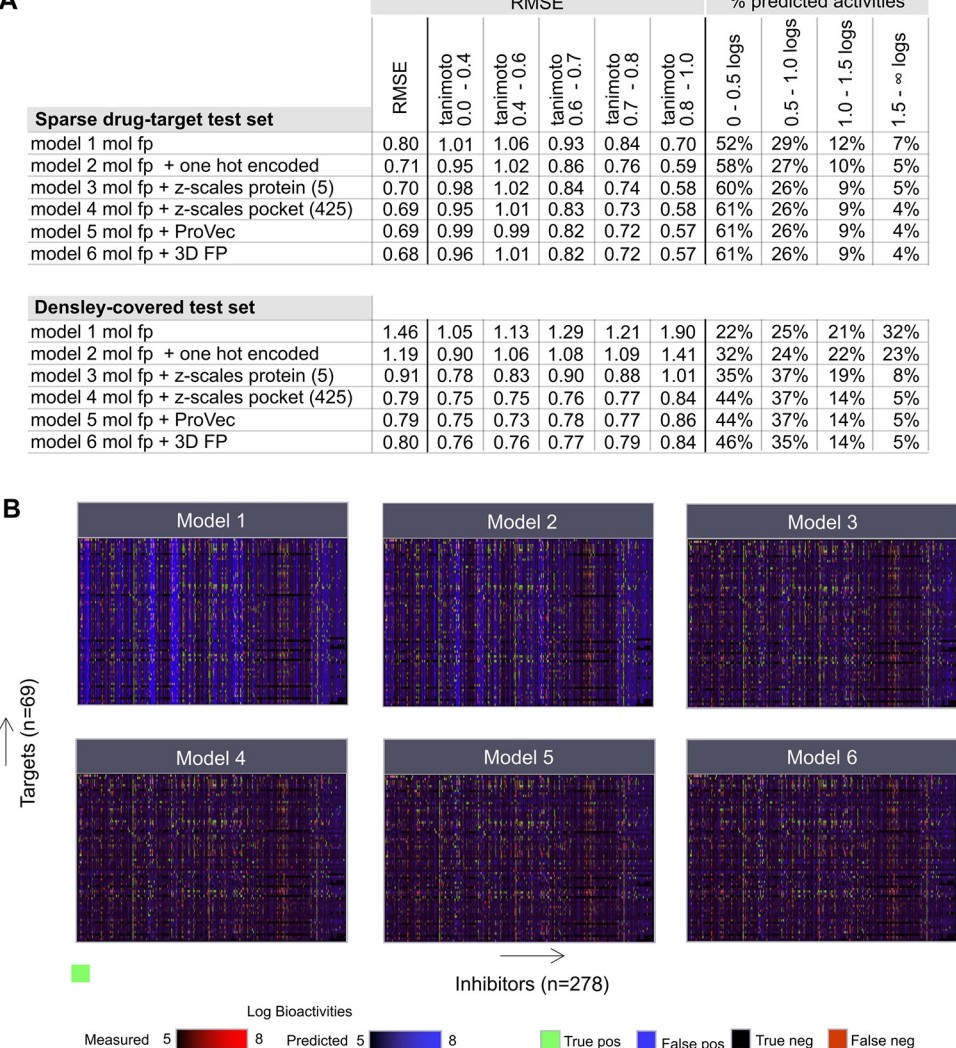

**Fig 4. Advanced protein features enhance the model's performance on top of drug fingerprints. (A)** The root-mean-square error (RMSE) performance of the six models on the 2 test sets. The table also includes the RMSE values calculated per subset (5 subsets defined based on the Tanimoto similarity of the molecular fingerprints) and the fraction of all data predicted accurately within 0–0.5 log unit, 0.5–1 log unit, 1–1.5, and higher than 1.5 log units. Model 1 comprises only of Morgan fingerprints (without any protein features), model 2 comprises of Morgan fingerprints (fp) and one-hot-encoded protein features, models 3 and 4 comprise of Morgan fp and z-scales, model 5 comprises of Morgan fp and ProtVec and model 6 comprises of Morgan fp and 3D convolutional fingerprints (3D FP). **(B)** The models' predictions for the densely-covered test set compared to the true labels. Black indicates matching true negatives, and green indicates matching true positives.

As expected, the RF model 1 with only Morgan features scored poorly on the densely-covered test set (RMSE = 1.46). This model used only compound fingerprints; thus, it is impossible to differentiate between the kinases. This is crucial as each compound in the dense set is tested on at least 60 kinases. Adding the one-hot-encoded kinase features showed improvement by lowering the RMSE to 1.19 (model 2). This indicates that even simple feature like one-hot-encoded provides some power to the algorithm to better associate chemical moieties with particular kinase, thereby improving performance. Swapping the one-hot-encoded features with more advanced features like z-scales lowered the RMSE further by 0.28 (model 3). Z-scales per binding site residue (425), ProtVec, and 3D FP scored similarly (RMSE = ~0.79).

Providing more detail in the protein features benefits the performance of the models. However, as the dense set is much more imbalanced with a high percentage of low potency values, the medium RMSE of all 69 RMSEs is 0.815, which indicates that using the medium potency in the train set for a kinase as a constant to predict the values for that kinase in the dense test set would perform reasonably for most kinases (**S4 Table**).

Interestingly, on average, the RMSE values drop in the sparse drug-target test set when the Tanimoto similarity (also known as Jaccard similarity) of the Morgan fingerprints (radius 2, 1024 bits) increases between the compounds of the train and test sets (**Fig 4A**). This is different for the densely-covered test set. Testing model 1 (Morgan fp) on the Tanimoto 0.8–1.0 similarity subset of the densely-covered test set shows increased RMSE compared to the other Tanimoto similarity subsets. This is also observed in model 2 with one-hot-encoded kinases as protein features. The differences between the Tanimoto similarity subsets decrease with z-scales calculated for the whole protein and decrease even further with the more advanced features like ProtVec and 3D FP (**Fig 4A**). Also notable are the differences between the sparse and densely-covered test sets regarding percentage data points correctly predicted within a specific log activity range. All models tested on the sparse test set show the same trend, i.e., most predictions fall within 0 and 0.5 log unit differences (between prediction and observation), and the more this range increases, the lower the percentages. Models 1 and 2 show a uniform distribution in the densely-covered test set. Models 4, 5, and 6 can double the percentage of correctly predicted data points within 0–0.5 log units compared to model 1.

Analysis of the predictions of the models on the densely-covered test set reveals that the models with more advanced protein features (z-scales, ProtVec, and 3D FP) are trying to minimize the false positive (FP) rate (**Fig 4B** and **S5 Table**). The threshold for calculating true and false positives was set at –log [activity] = 7. Models 1 and 2 show, in many cases, a high FP rate compared to the rest of the models. Unfortunately, models 3–6 have difficulties with recognizing the true positives. The true positive rate (TPR) for models 3–6 on the densely-covered test set is between 21% and 23% (**S5 Table**). The TPR slightly decreases (12–15%) when calculated per kinase (**S6 Table**). Each inhibitor in the dense test set is tested on at least 60 kinases, and typically among those, 2 to 5 kinases would show potency higher or equal to 7.0 (the defined threshold for true positive). Furthermore, we analyzed the median error in predicting the potency of each inhibitor (**S7 Table**). On the sparse drug-target test set, models 3–6 achieve ~15% TPR (**S8 Table**). The balanced accuracies and F1 scores using the same threshold of -log [activity] = 7 are provided in **S9 Table**.

## 3 Discussion

### The added value of protein features in predicting kinase activity profiles

The choice of test sets greatly influences the performances of the different models. The sparse drug-target set indicates how well the models can predict the activity of small molecules on a single protein kinase but falls short in accurately estimating model performance regarding kinome-wide profiles of small molecules. In our analysis, the compound features alone perform well on the sparse test set. The RMSE of model 1 using only Morgan features was evaluated at 0.8. One-hot-encoded targets lowered the RMSE to 0.71, and the advanced protein features (z-scales, ProtVec, and 3D FP) lowered the RMSE further to 0.69. However, the difference between the best and worse models was less than 0.15.

We built a densely-covered test set to more accurately evaluate the performance of the different models in predicting the activity of small molecules on multiple kinases. All 278 small molecules were tested on at least 60 kinases in this set. Here, model 1 scored poorly (RMSE = 1.46) as compound features alone are not enough to predict the binding profiles of

small molecules. Adding one-hot-encoded representations of the targets lowered the RMSE by ~0.3. The five z-scales generated with the KLIFS binding site sequences provided additional information (compared to one-hot-encoding) and lowered the RMSE by another ~0.3. Advanced features such as ProtVec and 3D FP lowered the RMSE by 0.66 compared to model 1. The difference between the best and worse models (1.46–0.79 = 0.67) was ~5.5 times higher than in the sparse drug-target set (0.8–0.68 = 0.12). Also, the RMSEs among models with the same features differed depending on the test set used. For example, model 6 tested on the densely-covered test set scored better (RMSE = 0.68) than when tested on the sparse drug-target set (RMSE = 0.8).

## Advantages and disadvantages of 3D convolutional fingerprints

The layers of 3D Convolutional Neural Networks can efficiently learn structural features from the provided protein structures, enabling them to distinguish a given kinase from all of the rest. This approach allows the algorithm to learn 3D features directly from the provided 3D biomolecular structures encapsulating the essence of a kinase that makes it unique from the rest. The authors believe the rapid increase of structural data would enable 3D convolutional networks to outperform sequence-based methods. Also, molecular dynamics (MD) simulations could contribute to the performance of the 3D CNN models as they can generate multiple and diverse snapshots from a single (starting) structure and, by doing so, enlarge not only the size of the data but also conformational space and flexibility. Moreover, advanced machine learning approaches like AlphaFold, [84] capable of accurately predicting the 3D structure of proteins from their amino acid sequences, could expand the coverage of 3D-KINEssences and simultaneously improve performance. Of the 311 protein kinases with at least one solved structure, 69 could be utilized in our workflow.

In this work, we propose a new type of protein fingerprints, 3D convolutional fingerprints (3D FP), generated with 3D convolutional neural networks (3D CNN) using biomolecular protein kinase structures from KLIFS. The performance of the 3D FP was estimated at RMSEs equal to 0.68 and 0.8 for the sparse and densely-covered sets, respectively. Moreover, 3D FP was compared to 2 other types of protein features: z-scales and ProtVec, where 3D FP outperformed z-scales and performed equally to ProtVec, indicating their applicability in machine learning pipelines to predict bioactivities of small molecules.

# 4 Online Methods

## Protein features

The kinases and their structural annotations were obtained from KLIFS through the KNIME analytic platform v4.1.4 with the KLIFS nodes inside the 3D-e-Chem extension. Kinases with at least ten solved crystal structures were selected. Structures that share identical orthosteric ligands in the same DFG conformation for a given kinase were removed from the data (only one representative structure was kept). This step ensures that the data provided to the convolutional neural network (CNN) does not contain repeats, as different ligands will introduce modifications in the structures. Structures with identical orthosteric ligands for a particular kinase were kept in different DFG conformations (e.g., DFG-in and DFG-out(-like)). The total number of selected kinases was 69. In total, 3400 structures were selected. The structures were prepared with Maestro's Protein Preparation wizard in combination with Prime to fill the missing side chains and Epik to protonate the ligands. See the next section, **3D convolutional fingerprints,** for the exact steps to prepare the 3D FPs. The kinase sequences were obtained from KLIFS [76] and UniProt, [85] respectively, and used as input for ProtVec and the z-scales. In house python script was written to generate the ProtVec vectors. The z-scales

(whole sequence and per residue) were prepared using the zScales from the CRAN Peptides package [86].

The 69 chosen kinases ordered alphabetically: ABL1, ACVR1, AKT1, ALK, AURKA, BRAF, BTK, CDK2, CDK8, CDK9, CHEK1, CHEK2, CLK1, CSNK1D, CSNK2A1, CSNK2A2, DAPK1, DDR1, DYRK1A, DYRK2, EGFR, EPHA2, EPHA3, ERN1, FGFR1, FGFR2, FGFR4, GSK3B, HCK, IGF1R, INSR, IRAK4, ITK, KDR, KIT, LCK, MAP2K1, MAP2K7, MAP3K5, MAP3K7, MAPK1, MAPK10, MAPK14, MAPK8, MAPKAPK2, MELK, MERTK, MET, NEK2, NTRK1, PAK1, PAK4, PDPK1, PIK3CA, PIK3CG, PIM1, PRKACA, PTK2, RET, RIPK2, ROCK1, RPS6KB1, SRC, STK24, SYK, TBK1, TGFBR1, TTK, and WEE1.

## 3D convolutional fingerprints

The 3D convolutional neural network (3D CNN) aims to learn essential features directly from the provided 3D biomolecular structures that distinguish the individual kinases. The 3D convolutional neural networks (3D CNN) were built using libmolgrid v 0.5.2 and PyTorch v1.13.1 [87]. Rather than training the model from scratch, we started with a pre-trained 3D CNN model by Koes et al. 2018 (Default2018) and further trained it to recognize the chosen kinases (transfer learning). 3D CNN consists of four 3D convolutional layers with kernel sizes (3, 3, 3), two 3D convolutional layers with kernel sizes (1, 1, 1), four average pooling layers, flatten layer, and an output layer (**Box 1**). The output layer has 69 units, equal to the number of chosen kinases (see section protein features in the methods). Libmolgrid randomly rotates the provided structures and generates the grids for the 3D convolutional and average pooling layers. The random rotation ensures that the model does not solely focus on a single location in 3D to learn features. The dimensions of the input grid for the first 3D layer were set at (14, 48, 48, 48). The network used the ReLU activation function. The 3D CNN used the stochastic gradient descent (SGD) optimizer with a learning rate set to 0.01 and momentum set to 0.9. Cross entropy loss was chosen to calculate the loss. The flatten layer "flattens" the learned representations of the convolutional layers (from 3D to 1D) while keeping the spatial information. This flattened output represents the learned 3D FP, comprising 216 decimal numbers. Once the 3D CNN model was trained and evaluated, the 3D FP features were generated by providing a single structure per kinase from the test set to 3D CNN and saving the activation of flatten layer.

---

### Box 1. The architecture of the 3D CNN used to generate the 3D FP

Sequential(

(unit1_pool): AvgPool3d(kernel_size = 2, stride = 2, padding = 0)

(unit1_conv): Conv3d(14, 32, kernel_size = (3, 3, 3), stride = (1, 1, 1), padding = (1, 1, 1))

(unit1_func): ReLU()

(unit2_conv): Conv3d(32, 32, kernel_size = (1, 1, 1), stride = (1, 1, 1))

(unit2_func): ReLU()

(unit3_pool): AvgPool3d(kernel_size = 2, stride = 2, padding = 0)

(unit3_conv): Conv3d(32, 64, kernel_size = (3, 3, 3), stride = (1, 1, 1), padding = (1, 1, 1))

(unit3_func): ReLU()

---

(unit4_conv): Conv3d(64, 64, kernel_size = (1, 1, 1), stride = (1, 1, 1))

(unit4 func): ReLU()

(unit5_pool): AvgPool3d(kernel_size = 2, stride = 2, padding = 0)

(unit5_conv): Conv3d(64, 128, kernel_size = (3, 3, 3), stride = (1, 1, 1), padding = (1, 1, 1))

(unit5_func): ReLU()

(unit6_pool): AvgPool3d(kernel_size = 2, stride = 2, padding = 0)

(unit6_conv): Conv3d(128, 8, kernel_size = (3, 3, 3), stride = (1, 1, 1), padding = (1, 1, 1))

(flatten): Flatten(start_dim = 1, end_dim = -1)

(output): Linear(in_features = 216, out_features = 69, bias = True)

## Compound features

The inhibitors from ChEMBL v31[73] and Christmann-Franck et al. 2016[38] were collected separately. In the case of Christmann-Franck et al., the smiles were extracted from the provided **S3 Table**. In the case of ChEMBL, the smiles were obtained by running an SQL query in the downloaded SQLite version of the database. The tautomers for the two sets were prepared using Ambit-Tautomer.[88] The two data sets were merged once prepared separately (see section **Bioactivity data**). The InChi keys, generated with Ambit-Tautomer, were used to create a kinase-inhibitor label for each activity value in the data sets to ensure duplicates were removed when merging. Inhibitors with molecular weights smaller than 180 or bigger than 700 were removed. Salts were stripped. The compound features consisted of 1024 bits of Morgan fingerprints with a radius set to 2 and were prepared using rdkit v2020.09.1.0 and python v3.7.

## Bioactivity labels

The bioactivity data of the following two resources were integrated and used as labels in the machine learning pipelines: ChEMBL v31 (20M bioactivities) and Christmann-Franck et al. 2016 (357K bioactivities). The data from the two datasets were prepared in the same way (**Box 2**). SQL queries were applied to filter the protein kinases from ChEMBL by using their UniProt IDs.

---

### Box 2. Integration of ChEMBL v31 and Christmann-Franck bioactivity datasets

1. The bioactivity data was filtered for the 69 kinases (see section **protein features**).

2. In ChEMBL, bioactivities with a confidence score $< 7$ were removed.

3. The following data types were selected: IC50, Ki, Kd, Potency, %inhibition and % activity.

---

4. Bioactivities without any value were removed.

5. The bioactivities with exact measures (i.e., relation "=") and type nM were converted to log activities.

6. The bioactivities with relation "> =" and value 10,000 with type nM were converted to -log[activity] = 5.

7. The bioactivities with relation "<" and type nM were removed.

8. The bioactivities of type %inhibition and %activity were converted to log activity of 5 if inhibition < 10% or activity > 90%. The percentage inhibition or activity was measured at 10,000 nM. The rest of the activities of these types were removed.

After step 8, all data were merged. See the section **Compound features** for the steps to prepare the compounds. The median log activity values were taken if a small molecule was measured on the same kinase multiple times. In total, 169,723 bioactivities and 69 unique kinases were used in the machine learning pipelines.

## 3D-KINEssence—machine learning workflow

The generation of the kinase 3D convolutional fingerprints in combination with compound fingerprints (Morgan fingerprints prepared with rdkit, radius equals two, and number of bits equal 1024) comprise the 3D-KINEssence pipeline. The random forest was built with scikit-learn v 0.22[89]. The data set consisted of 170K activity data. In the case of the sparse training and test sets, the split was generated using train_test_split from scikit-learn with a test size equal to 0.3 and stratify per kinase. In the case of the dense training and test sets, the split was done based on the number of kinases per compound. If a compound was tested on more than or equal to 60 kinases, it was included in the dense test set, otherwise, it was part of the dense training set. For both dense and sparse workflows: a single compound (regardless of how many kinases it was tested) can only be in the training or testing set (not both). On the other hand, all 69 kinases are present in both sets. The number of trees for each model was set to 100, while the rest of the settings were kept at default.

## Multi-stack visualization of performance

We created heatmaps showing the overlap between measured bioactivities (in blue) versus predicted values from the respective models 1–6 (shown in red). To distinguish true positives, the light pink color (ff66cc) representative of a matching true and predicted value, was replaced by a bright green (66ff66) color using photoshop (Adobe Photoshop Version: 19.0 20171103. r.190). The green color was set to 100% saturation and +1 lightness.

## Supporting information

**S1 Fig. Skewness plots of both test sets.** A) The sparse test set. B) The dense test set. The dense test set is skewed to the left.
(TIF)

**S2 Fig. Kurtosis plot of the 69 protein kinases used in the machine learning pipeline.** The plot was generated with the kurtosis function (using Fisher's definition) of pandas 1.3.5.
(TIF)

**S1 Table. The kinases used in the machine learning pipeline and their groups classification (TK, TKL, STE, CK1, AGC, CAMK, CMGC, Other and Atypical).**
(XLSX)

**S2 Table. The PDB structures used to train and test the 3D Convolutional Neural Networks model and generate 3D FP.**
(XLSX)

**S3 Table. The 278 inhibitors of the dense set.**
(XLSX)

**S4 Table. Hypothetical RMSE and R2 values per kinase: the predicted potencies for a kinase is constant value and equal to the median potency of that kinase in the training set.**
(XLSX)

**S5 Table. The true positives (TP), false positives (FP), true negatives (TN) and false negatives (FN) for each compound in the densely-covered test set.** The tabke also includes the true positive rate (TPR) and false positive rate (FPR).
(XLSX)

**S6 Table. The true positives (TP), false positives (FP), true negatives (TN) and false negatives (FN) for each kinase in the densely-covered test set.** The tabke also includes the true positive rate (TPR) and false positive rate (FPR).
(XLSX)

**S7 Table. Upper table: The median value of absolute predicted minus obseved log activity for each of the 278 compounds in the densely-covered test set.** The predictions of each compound are generated with models 1 to 6. Lower table: the median of model 6 minus the medians of the other models.
(XLSX)

**S8 Table. The true positives (TP), false positives (FP), true negatives (TN) and false negatives (FN) for the sparse test set.** The tabke also includes the true positive rate (TPR) and false positive rate (FPR).
(XLSX)

**S9 Table. The balanced accuracies and F1 scores for the sparse and densely-covered test sets.**
(XLSX)

## Author Contributions

**Conceptualization:** Georgi K. Kanev, Bart A. Westerman.

**Data curation:** Georgi K. Kanev, Yaran Zhang.

**Formal analysis:** Georgi K. Kanev, Yaran Zhang, Iwan J. P. de Esch.

**Funding acquisition:** Georgi K. Kanev, David Bailey, Thomas Würdinger, Chris de Graaf, Bart A. Westerman.

**Investigation:** Georgi K. Kanev, Yaran Zhang, Bart A. Westerman.

**Methodology:** Georgi K. Kanev, Yaran Zhang, Bart A. Westerman.

**Project administration:** Thomas Würdinger, Iwan J. P. de Esch, Bart A. Westerman.

**Resources:** Albert J. Kooistra.

**Software:** Georgi K. Kanev.

**Supervision:** David Bailey, Thomas Würdinger, Chris de Graaf, Iwan J. P. de Esch, Bart A. Westerman.

**Validation:** Georgi K. Kanev, Yaran Zhang, Bart A. Westerman.

**Visualization:** Georgi K. Kanev.

**Writing – original draft:** Georgi K. Kanev, Bart A. Westerman.

**Writing – review & editing:** Georgi K. Kanev, Yaran Zhang, Albert J. Kooistra, Andreas Bender, Rob Leurs, David Bailey, Thomas Würdinger, Chris de Graaf, Iwan J. P. de Esch, Bart A. Westerman.

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
