## [Decision Letter · Decision Letter 0]

21 Oct 2022

Dear Dr. Westerman,

Thank you very much for submitting your manuscript "Predicting the target-landscape of kinase-inhibitors using 3D convolutional neural networks on densely covered chemogenomic data" for consideration at PLOS Computational Biology.

As with all papers reviewed by the journal, your manuscript was reviewed by members of the editorial board and by several independent reviewers. In light of the reviews (below this email), we would like to invite the resubmission of a significantly-revised version that takes into account the reviewers' comments.

Specifically, the manuscript lacks essential information and reveals weaknesses in the method, which is an issue for a paper required to propose an outstanding method of exceptional importance and wide usage. Incidentally, code availability was not declared. Whether a major revision may address the fundamental problems is uncertain.

We cannot make any decision about publication until we have seen the revised manuscript and your response to the reviewers' comments. Your revised manuscript is also likely to be sent to reviewers for further evaluation.

Sincerely,

Francesca Fanelli

Guest Editor

PLOS Computational Biology

Nir Ben-Tal

Section Editor

PLOS Computational Biology

Reviewer's Responses to Questions

**Comments to the Authors:**

Reviewer #1: In the paper entitled Predicting the target-landscape of kinase-inhibitors using 3D convolutional neural networks on densely covered chemogenomic data by Kanev et al. the authors devise and utilise a structurally derived fingerprint (FP) for the kinase domain of protein kinases, and compare the performance of their FP to other literature methods. They show performance on two literature derived datasets, one sparse, one dense, and conclude that their FP equals or slightly outperforms literature standards, depending on the dataset. The paper has a clear build-up, is well illustrated, but experimentally has serious shortcomings and lacks sufficient detail in the computational approach taken.

Major textual and conceptual problems:

- The mention of personalised medicine and how this research supports this is poorly substantiated. Personalised medicine implies specific mutations or specific deregulations in a cancer, where this new FP might mostly give a clearer image in polypharmacology applications.

- From the introduction, it is unclear what the goal of this paper is. The fingerprint? The pipeline? The predicted affinity matrix? The introduction clearly lists a large amount of preceding papers that covered some (combination of) similar goals, yet the authors do not compare the performance of their models with any literature (CNN) model. The results shown, and the discussion thereof, does not extend at all beyond their own model and dataset. Literature model performance should at least be compared to show the merit of this new approach.

- From the text, it is unclear what the Random Forest models were tasked to predict. Talking about RMSE seems to imply that the RF model predicts -log[XC50]. This should be made way more obvious in the text. If this is indeed the case, correlation (R2) of predicted and observed values should be discussed and shown.

- Figure 4, and the supplementary tables, talk about true/false positives, but it is unclear what threshold value was set for a compound to qualify as ‘active’.

- The method of constructing a FP based on a CNN per protein is highly debatable. This strategy could be a major reason why many structures were needed per protein, as the resulting classification task was otherwise extremely challenging due to the imbalance of the underlying dataset. Variational Auto Encoders have previously been utilised for chemicals to generate an embedded latent space that is useable as FP for machine learning tasks (e.g. Winter et al. https://pubs.rsc.org/en/content/articlehtml/2019/sc/c8sc04175j), a more similar strategy might have worked here, also leading to a more general applicable FP that does not require the construction and training of a dedicated CNN per protein.

- The RF models for the dense dataset drop an incredible 0.48 in RMSE (1.67->1.19) by the addition of only 5 features in the z-scales whole sequence models. This dramatic effect of 5 features warrants more discussion.

- The size of the dense dataset (155 compounds) is definitely on the small side, yet no comment is made on how this could impact model performance at all, instead the lack of performance is wholly blamed on the lack of target information (which apparently wasn’t a problem in the sparse dataset).

- Practically, one could definitely argue that an RMSE of 0.82, 0.84, 0.87 or 0.92 is roughly equal in its reliability, showing that the addition of any protein data at all only very slightly improves model performance.

- It should be considered how the skewedness of the affinity values varies between the two datasets, and how this ties into model performance. Does the RF model just predict the median for that particular kinase in the dense dataset?

Minor textual points:

- The cost of developing has increased 9-fold, 9 times implies 9 specific price hikes.

- Swapping Figure 1B and 1A helps to refer in the order mentioned in the text. Reason for having current Figure 1A is unclear, apart from mentioning that there are trends in conservation nothing is done with this information.

- In line 30, the reference 20 might be updated to the ’22 update, which also increases the 50 approved inhibitors to 68.

- From the text, it is unclear whether the full protein sequence was used, or the kinase domain sequence.

- In line 133 the 1.4% disagrees with the 2.4% coverage mentioned earlier in the text.

- Figure 3 is said to show diversity of molecules (line 135), but only shows the number of inhibitors.

- Figure 3B shows ‘log activity’, where -log[inhibitor] is presumably meant.

- From the text (line 154) and Figure 4 it is unclear whether Tanimoto distance or similarity have been used. Based on the results, the similarity makes sense, but ‘score’ is not obvious.

- No details at all have been given on the tSNE embedding generation. Hyperparameters, tSNE implementation etc. should be given to make the graph interpretable.

Reviewer #2: The manuscript (ID: PCOMPBIOL-D-22-01124) entitled “Predicting the target-landscape of kinase-inhibitors using 3D convolutional neural networks on densely covered chemogenomic data” reports the development of a machine learning pipeline called 3DKINEssence, which is based on a novel type of structural fingerprints generated through 3D convolutional neural networks. The idea behind the study can be of high interest in modern drug discovery, especially in the contexts of kinase polypharmacology design and personalized medicine. However, the manuscript presents several issues that should be addressed.

• A more thoroughly discussion on the concepts behind the generated 3D-CNN kinase fingerprints and on its main differences with respect to the other approaches encoding protein features should be provided. Perhaps, adding explanatory examples related to one or more compounds would help to better understand how the structural features of the proteins are perceived by the different approaches and the opportunities offered by the newly developed pipeline.

• The following sentences should be clarified to avoid potential misunderstandings: (i) page 5, lines 78-81; (ii) page 6, lines 105-107.

• The data activity skewness related to the kinases considered in the study was briefly discussed in the text and depicted in the right side of Figure 1A. A comment also on the kurtosis of the activities would help to better understand the data distribution across the different kinase datasets.

• The full list of crystal structures used for the generation of the 3D FP should be provided as a supplementary file.

• Figure 1 should be revised, as it presents a different number of panels with respect to those referenced in the text and in the corresponding caption. Moreover, the panels in Figure 1 should also be reorganized to make their reference in line with the text (for example, panel B now is referenced before panel A in the text). The type of molecular fingerprints employed for the generation of the tSNE plot should be clarified in the Figure 1 caption.

• Supplementary Table 1 should also provide information on the 9 groups to which the kinase proteins were assigned to, to better support the sentence at page 5, lines 77-78. Moreover, a reference to the Supplementary Table 2 is missing in the text.

• The manuscript presents some typos and grammatical issues. I would suggest performing a careful revision of the text.

• The bibliography is not exhaustive, albeit being already extensive. Few examples that would need to be referenced at their first occurrence in the text are: (i) “z-scales and ProtVec” at page 4, line 65; (ii) “Kieo database” at page 5, line 73; (iii) “Morgan fingerprints” at page 6, line 95; (iv) “KNIME analytical platform” at page 6 line 103; (v) “one-hot encoding” at page 7 line 119; (vi) “RDKit” at page 7, line 121; (vii) “UniProt” at page 19, line 508; (iix) “zScales CRAN package” at page 19, line 511; (ix) “PyTorch” at page 19, line 520; (x) “scikit-learn” at page 21, line 547.

• A more detailed description on the activity data filtering should be provided. This with particular regards to the type of assays considered for the activity data and how the presence of potential false positives records was managed.

Reviewer #3: The authors present a machine learning model using random forest (rf) regression for the prediction of activities of ligands for a set of 71 kinases. To this end, the binding pockets of kinases are encoded as 3D fingerprints. Those fingerprint together with ECFP4 fingeprints of ligands form the input of the rf model to predict potency of a ligand vs. the target encoded. The main novelty of their work stems from the construction of the 3D fingerprint using 3D convolutional neural network models. The manuscript is well written and structured making it easy to follow. This is aided by the clear and informative figures. However crucial information is missing from the manuscript that would prevent publication in its current form:

Random forest models are built based on data extracted from ChEMBL and other sources. These methods are then tested on a sparse and dense data set. These data sets and their sources are not described in the manuscript. What are the activity annotations of the test sets (Kd, IC50, ...)? It also needs to be assured that this (or part of this) data is not also included in the training data. A meaningful evaluation of the results would only be possible if the independence of the test and training data sets can be assured.

P3L25,P6L90: should refer to Figure 1C instead of 1B

P6L94: should refer to Figure 1D instead of 1C

P6L98: should refer to Figure 1E instead of 1D

P6L109ff: Doing the math, the CNN models are trained and tested on highly unbalanced data. Assuming the minimum of 20 structures/kinase. The positive training set would consist on the order of 10 samples (50%) and the negative training set on the order of 70*10=700 samples from the other kinases. The reported performance would then be reported on the test set (I assume) of ca. 5 postive and 350 negative samples (25%). In this case precision and recall (and maybe F1-score as a combined measure) are indeed meaningfult, however accuracy is not. A model only predicting negative would already have an accuracy of ca. 0.985. As a combined score, either the balanced accuracy or a score like F1 could be reported instead.

P19L25: The authors should provide a more details about the training and evaluation method, e.g., what are the absolute sizes of the training/test sets per target? (the supplemented contains the positive training set size, the sizes of the negatives sets could be included there per target); which kind of validation was done for hyperparameter optimization?; did the authors perform repeated trials?; and if so how were the final models for fingerprint generation chosen?

P6L111: Seems to refer to Figure 2 instead of Figure 3

P7L139-140: "The root-mean-square-error ...": I feel the context of this statement is missing, to which test sets do the authors refer?

P7L139ff: To put the RSME results into persepective, the authors should provide the RSMEs for two baseline models:

1) A model that always predicts the same value, namely the average potency of tre training set compounds

2) A model that always predicts the same value per target, namely the average potency of the training set compounds for that target

P8L153: How was the Tanimoto similarity determined? Was the most similar compound of the training set or some kind of average taken?

P9L168/Fig4B: How were True/False positives/negatives defined given that the models are regression models?

P17L459ff: Legend Figure1: The legend combines subfigures A and B as a single subfigure A. Legends/references in the text should be fixed. (The font of legend 1 does not match the font of the other figure legends.)

P19L512: chose -> chosen

P20Box 1: A lot of different activity types are mixed in the data preparation step: IC50, XC50, EC50, AC50, Ki, Kd, Potency the quantitative values will depend on assay conditions and are not comparable directly introducing a significant amount of error in the initial data. This issue needs to be addressed and/or discussed in some form in the manuscript. The authors should also investigate whether a model trained on a smaller training set of higher quality (restricting annotations to Ki/Kd (and IC50) would yield better results.

P21L554: B.W -> B.A.W.

Figure 3B: Sparse data set: The figure is informative in the sense that it gives a good impression of the variance in the number of molecules per target. To this end, molecules were sorted according to their target from top to bottom along the x-axis. Given that the x-axis represents 8950 molecules, individual pixels shown cannot represent individual protein-ligand interactions. I would suggest, for the molecules within each target, these should be sorted in order of increasing (or decreasing) activity, so that the potency range for each target becomes visible.

**Have the authors made all data and (if applicable) computational code underlying the findings in their manuscript fully available?**

Reviewer #1: **No: **The supplementary tables detail median values, but separate values are not available. Code availability is not detailed in the paper.

Reviewer #2: None

Reviewer #3: **No: **No code has been provided. The activity annotations and structures of ligand of training and test set are not available.

PLOS authors have the option to publish the peer review history of their article (what does this mean?). If published, this will include your full peer review and any attached files.

Reviewer #1: No

Reviewer #2: No

Reviewer #3: **Yes: **Martin Vogt
---

## [Decision Letter · Decision Letter 1]

6 Apr 2023

Dear Dr. Westerman,

Thank you very much for submitting your manuscript "Predicting the target-landscape of kinase-inhibitors using 3D convolutional neural networks on densely covered chemogenomic data" for consideration at PLOS Computational Biology.

The revised manuscript has been re-evaluated by the three original Reviewers who found significant improvements in the text. However, a number of points still need to be addressed for the manuscript to be published (please, see the Reviewer comments). Moreover, in the revised version you promised to provide all data and scripts through a github link. Please, do it prior to acceptance for publication.

Sincerely,

Francesca Fanelli

Guest Editor

PLOS Computational Biology

Nir Ben-Tal

Section Editor

PLOS Computational Biology

Reviewer's Responses to Questions

**Comments to the Authors:**

Reviewer #1: Review comments are uploaded as an attachment

Reviewer #2: The revised version of the manuscript (ID: PCOMPBIOL-D-22-01124R1) presents substantial improvements. However, it still presents few issues that should be addressed in my opinion.

• Figure 1B should be revised to help better understanding which regions of the kinases binding site are likely to present higher differences in terms of amino acids composition. Perhaps, the substitution of Figure 1B with a 3D representation of the protein binding site highlighting less conserved regions would also help better understanding how this information could be exploited by the convolutional neural networks when trained on the 3D biomolecular structural data.

• The sentence at lines 99-101 should be clarified to avoid potential misunderstandings. Moreover, Supplementary Figure 2 should also be revised considering that some of the negative values in it are out of scale (Y-axis).

• The sentence at lines 32-33 should be clarified. LigPrep is a tool available in Maestro (Schrödinger), developed for the preparation of ligands; it seems that it was used for the optimisation of the ligand-protein complexes. Moreover, further considerations should also be provided with respect to the protonation and tautomeric states potentially accessible by kinase modulators in the datasets, which seem to be not evaluated.

• Compounds with reported IC50, Ki, Kd, Potency, % of inhibition and % of activity were considered in the study (see step 3 in Box1). The activity threshold to which the % of inhibition and % of activity have been evaluated for the compounds according to literature data should be reported. Moreover, additional details on how, for example, duplicate records were removed during the dataset preparation should also be described. This considering that the % of inhibition evaluated at 1uM might significantly differ with respect to those observed at 20uM, 50uM or 100uM, and that the same compound might have been reported significantly different activity values across different studies.

• Supplementary tables should be re-organised to make their reference in line with the text (for example, Table S4 is referenced before Table S3). Moreover, references to Supplementary Tables 2 and 5 seem to be missing in the text.

• The acronym of T-SNE should be spelled out at its first occurrence.

• References to relevant literature studies and to the software employed should be placed at their first occurrence in the text. Moreover, the manuscript presents several typos that need to be removed (few examples are reported in the lines 133, 539, 549 and in the caption of Figure S2).

Reviewer #3: The authors have submitted a first major revision of their original manuscript

"Predicting the target-landscape of kinase-inhibitors using 3D convolutional neural

networks on densely covered chemogenomic data."

While the general approach has remained unchanged, the authors have changed the architecture of their 3D-CNN for the construction of protein 3D Kinases FPs. This modification significantly changes the generation of 3D-FPs, but it seems to be a meaningful modification over the original approach. Overall the manuscript has improved significantly, however some revisions are still required. While the written English is understandable, it could still improve from some editing, especially in the results section where it is on first reading often ambiguous whether given RMSE values refer to the performance of a model or the difference in performance between two models.

Points:

L20: "increased 9 times" -> "increased by a factor of 9"

L46: "random forest" -> random forests

L88: "large part" -> "a large part"

L91: "as result end in" -> "as a result ends in"

L92: "that kinase have " -> "kinases generally have"

L99: "than the sparse test set" -> "compared to the sparse set"

L100 -> "that most kinases are close to 0 " -> "it being close to 0 for most kinases"

L123/L546ff: According to manuscript of Fig. 2, the 3D-FP corresponds to the flatten layer. However, the size of the flatten layer and thus the 3D FP is not explicitly mentioned. This should be added here and/or in the main text (L123)

L140ff/L572ff: The manuscript is still a bit unclear what training and test sets where chosen and how training was done. Were different models trained separately for the dense and the sparse data set? Was part of the data used for training and another for testing or was data from one source used for training and then tested on data from another source. The machine learning workflow should be extended to include this information in the methods and/or L140

L180: The threshold potency values for positive and negative predictions should be explicitly stated in the manuscript and not only the reviewer responses.

Original points by the reviewer that were not/inadequately addressed:

"""

P6L109ff: Doing the math, the CNN models are trained and tested on highly unbalanced data. Assuming

the minimum of 20 structures/kinase. The positive training set would consist on the order of 10 samples

(50%) and the negative training set on the order of 70*10=700 samples from the other kinases. The

reported performance would then be reported on the test set (I assume) of ca. 5 positive and 350 negative

samples (25%). In this case precision and recall (and maybe F1-score as a combined measure) are indeed

meaningfult, however accuracy is not. A model only predicting negative would already have an accuracy

of ca. 0.985. As a combined score, either the balanced accuracy or a score like F1 could be reported

instead.

Response: Thank you, F1 and balanced accuracy has now been calculated as well. The convolutional neural

network part has now been revised to only one model. A single 3D CNN was in this newer version trained on all

structural data to learn and generate the 3D FP. However, the issue you address (unbalanced data) still remains

and has been discussed in the text.

"""

Comment: I could not find the discussion but the point should be moot due to the changed architecture of the CNN. However, The SI contains balanced accuracy and F1 scores for the RF models, which is appreciated.

"""

P7L139ff: To put the RSME results into persepective, the authors should provide the RSMEs for two

baseline models:

1) A model that always predicts the same value, namely the average potency of tre training set

compounds

2) A model that always predicts the same value per target, namely the average potency of the training set

compounds for that target.

Response: These are very good points, thank you. We have incorporated them in the discussion.

"""

Comment: The values are report in the supplemental information, but I could not find a discussion of them in the manuscript.

**Have the authors made all data and (if applicable) computational code underlying the findings in their manuscript fully available?**

Reviewer #1: **No: **No code or code availability statement provided

Reviewer #2: None

Reviewer #3: **No: **Code has not been made available. Curated kinase activity data is not provided.

PLOS authors have the option to publish the peer review history of their article (what does this mean?). If published, this will include your full peer review and any attached files.

Reviewer #1: No

Reviewer #2: No

Reviewer #3: No

Figure Files:

Data Requirements:

Reproducibility:

References:

---

## [Editor Report · Decision Letter 2]

25 Jun 2023

Dear Dr. Westerman,

We are pleased to inform you that your manuscript 'Predicting the target-landscape of kinase-inhibitors using 3D convolutional neural networks' has been provisionally accepted for publication in PLOS Computational Biology.

Best regards,

Francesca Fanelli

Guest Editor

PLOS Computational Biology

Nir Ben-Tal

Section Editor

PLOS Computational Biology

---

## [Editor Report · Acceptance letter]

25 Jul 2023

PCOMPBIOL-D-22-01124R2 

Predicting the target-landscape of kinase-inhibitors using 3D convolutional neural networks

Dear Dr Westerman,

I am pleased to inform you that your manuscript has been formally accepted for publication in PLOS Computational Biology. Your manuscript is now with our production department and you will be notified of the publication date in due course.

With kind regards,

Zsofi Zombor
